# Video Anomaly Detection Based on Convolutional Recurrent AutoEncoder

**DOI:** 10.3390/s22124647

**Published:** 2022-06-20

**Authors:** Bokun Wang, Caiqian Yang

**Affiliations:** 1College of Civil Engineering and Mechanics, Xiangtan University, Xiangtan 411100, China; 201921002175@smail.xtu.edu.cn; 2School of Civil Engineering, Southeast University, Nanjing 210096, China

**Keywords:** video anomaly detection, deep learning, convolutional long–short-term memory, convolutional autoencoder

## Abstract

As an essential task in computer vision, video anomaly detection technology is used in video surveillance, scene understanding, road traffic analysis and other fields. However, the definition of anomaly, scene change and complex background present great challenges for video anomaly detection tasks. The insight that motivates this study is that the reconstruction error for normal samples would be lower since they are closer to the training data, while the anomalies could not be reconstructed well. In this paper, we proposed a Convolutional Recurrent AutoEncoder (CR-AE), which combines an attention-based Convolutional Long–Short-Term Memory (ConvLSTM) network and a Convolutional AutoEncoder. The ConvLSTM network and the Convolutional AutoEncoder could capture the irregularity of the temporal pattern and spatial irregularity, respectively. The attention mechanism was used to obtain the current output characteristics from the hidden state of each Covn-LSTM layer. Then, a convolutional decoder was utilized to reconstruct the input video clip and the testing video clip with higher reconstruction error, which were further judged to be anomalies. The proposed method was tested on two popular benchmarks (UCSD ped2 Dataset and Avenue Dataset), and the experimental results demonstrated that CR-AE achieved 95.6% and 73.1% frame-level AUC on two public datasets, respectively.

## 1. Introduction

In order to improve the safety of people’s lives and public property, video surveillance systems have been widely installed in public places such as train stations, airports, hospitals, markets, schools, and resident centers. The main goal of social public safety risk prevention and control is to detect abnormal events accurately and timely. However, it is a tedious process to monitor the surveillance videos at a continuously faster rate, which leads to inefficient utilization of surveillance cameras and requires human presence for monitoring. Hence, video anomaly detection has recently become an important research problem in computer vision [1,2]. Given a surveillance video clip, the aim of frame-level video anomaly detection is to identify frames where there is an event or behavior that differs from the expectations or that appears infrequent. These abnormal events usually include fights, riots, violations of traffic rules, sdtrampling, holding arms, and abandoning luggage. However, video anomaly detection in general is a vast, crucial, and challenging research topic due to the ambiguity of anomaly definitions, the paucity of anomalous data, and the complex environmental background.

In general, current research work of video anomaly detection contains two procedures: feature extraction and model learning [3]. Feature extraction can be achieved by hand-crafteded feature technology or automatic feature extraction technology (features-based representation learning or deep learning). For the model learning procedure, normal samples are used for learning the detection model, and then, the testing samples that do not conform to the learned model are judged as abnormal events. There are three main categories of feature extraction approaches. (1) Trajectory-based methods [4]: Various methods track the target to obtain trajectory features and achieve satisfactory detection results for anomalies in both speed and direction, but target tracking in dense scenes is a big problem. For example, the authors in [5] studied the detection of abnormal vehicle trajectories, such as illegal U-turns. The authors in [6] extracted human skeleton trajectory patterns and were thus limited to detecting human behavioral anomalies. (2) Methods based on variable features [7,8]: Various methods take video frames as a whole and extract some simultaneous or mid-level features such as spatiotemporal gradients, histogram of gradient, optical flow, etc., which are effective in moderately crowded and dense environments. In [9], the authors proposed to associate the optical flows between multiple frames to capture short-term trajectories and to introduce the histogram-based shape descriptor to describe such short-term trajectories. (3) Grid feature-based method [10]: For the reason that each grid can be evaluated separately, this method often divides the video frame into multiple small grids through dense sampling, and then extracts overlapping features from the subdivided grids. As an example, Roshtkhari employed a probability density function to encode spatio-temporal configurations of video volumes based on spatio-temporal gradient features.

Furthermore, it could be divided into three categories by different model learning strategies. (1) Cluster-based methods [11]: these methods are often based on the hypotheses that normal samples belong to a category or are closer to one cluster center, while the abnormal samples do not belong to any category or away from any cluster center, and then the normal samples are clustered to build the detection model. In [12], the set of features generated by a convolutional autoencoder are clustered, and a one-versus-rest classifier is trained that discriminates between the clusters to detect the anomaly. (2) Sparse reconstruction based method [13]: This type of method assumes that the sparse linear combination of normal patterns can represent normal activities with the smallest reconstruction error, and because there is no abnormal activity in the training dataset, it can represent abnormal patterns with larger reconstruction errors. One such method is introduced by Hasan [14], where the use of combining 2D convolutions to autoencoders was produced, wherein the 2D convolutions were taken as input specific raw video segments. (3) The method based on the probability model [15]: This method considers that normal samples that conform to a certain probability distribution, while abnormal samples do not match this distribution. In [15], the detection of anomalies in a video is based on the hypothesis that the normal samples can be associated with at least one Gaussian component of a Gaussian Mixture Model (GMM), while anomalies do not belong to any Gaussian component.

Recently, the latest progress of deep learning has proven the obvious advantages of artificial intelligence-based methods and not be confined to many computer vision applications [16]. As one of the topic tasks in computer vision, video anomaly detection is no exception. Unlike the hand-crafteded feature-based methods, these deep learning-based methods often use pre-trained neural network architecture to extract high-level features, or build an end-to-end anomaly detection model with existing network architecture. For the latter idea [1,11,12,13,14,15,16,17,18,19,20,21,22,23], the feature extraction step and model building step are jointly optimized with one network. These end-to-end deep neural networks contain Deep Auto-Encoders (AE, Auto-Encoder) [14], Deep Siamese Networks (DSN) [17], and Generative Adversarial Nets (GAN) [18]. However, these network models are often designed for other tasks such as generative models, compression, etc., rather than for anomaly detection tasks.

Different from the possible solutions discussed earlier, in this paper, we propose a new deep learning-based method called the Convolutional Recurrent AutoEncoder (CR-AE) for video anomaly detection. Specifically, the proposed method is based on the combination of an attention-based Convolutional Long–Short-Term Memory (ConvLSTM) network and the convolutional encoder of the AutoEncoder, which are employed to capture the irregularity of the temporal pattern and spatial irregularity, respectively. Then, the convolutional decoder of the AutoEncoder is utilized to remodel the input video clip, and the reconstruction errors are further employed to detect abnormal frames. This is due to the reason that if the CR-AE has never observed a similar abnormal pattern before, it may not be able to reconstruct the input video clip well. Before our work, Hasan et al. [14] proposed to learn temporal regular patterns using a convolutional AutoEncoder with limited supervision to detect the video abnormal temporal events. Different from Hasan’s work, the proposed method could simultaneously detect the spatial and temporal anomalies. In addition, frame-level annotation is carried out on two public datasets called the UCSD ped2 dataset and the ShanghaiTech dataset to evaluate anomaly detection performance of our method. The experimental results demonstrate that our method has good characteristics of strong generalization ability and outperforms the state-of-the-art methods.

In summary, the main contributions of this study are as follows:We proposed an end-to-end deep learning framework for anomaly detection called Convolutional Recurrent AutoEncoder (CR-AE) for video anomaly detection. It is established by encoding the spatial regularity and temporal pattern with two common network architectures. They are the attention-based Convolutional Long–Short-Term Memory (ConvLSTM) network and the convolutional AutoEncoder (ConvAE). To the extent of our knowledge, this is the first time that the hybrid architectures of the attention-based ConvLSTM and ConvAE have been considered for video anomaly detection.We adopted only a network to simultaneously detect the spatial and temporal anomaly to replace the conventional two-stream network. Compared with the conventional two-stream network, the CR-AE need not extract optical flow and train the weights of the two architectures.We extensively evaluated our approach on the publicly available video anomaly detection datasets. The experiment demonstrates that our approach attains superior results compared to the state-of-the-art methods.


The remainder of this paper is structured as follows: Section 2 summarizes the related literature about existing anomaly detection. Section 3 describes the architecture of the proposed approach. Experimental evaluation on two public experiments is given in Section 4. Finally, Section 5 draws the conclusions of this paper.

## 2. Related Work

This section outlines the previous works on existing video anomaly detection methods, which include the hand-crafted feature-based and deep learning-based anomaly detection method.

### 2.1. Hand-Crafted Feature-Based Anomaly Detection Method

Early research on video anomaly detection adopted the hand-crafted features to represent the appearance and movement characteristics of pedestrians, and then machine learning method was used to learn the anomaly detection model. According to whether the object detection and object tracking procedure is adopted, these methods fall into two broad categories: anomaly detection methods based on trajectories and anomaly detection methods on cuboids.

Each trajectory represents the movement of a target as a sequence of image coordinates. The main idea of trajectory-based methods is based on the assumption that the anomalous trajectories of the abnormal events differ from the normal patterns. Junejo et al. [24] utilized the size, position, and speed of the trajectory as the feature to represent the event and to train a dynamic Bayesian network for abnormal behavior detection. Similarly, Kang et al. [25] proposed to utilize trajectory features to build a hidden Markov model to achieve an anomaly detection model. Similarly, Wang et al. [26] projected a dense trajectory algorithm that first densely sampled the feature points, then extracted the point trajectory features and encoded them, and then classified them through support vector machines in the end. After that, Wang improved the feature regularization and encoding method, and employed an improved method dense trajectory algorithm to represent the event in the video [27]. However, the detection result of the trajectory-based method depends on the accuracy of the object tracking method. Furthermore, the results of these methods degrade in the crowded or complex scenes where there is a lot of occlusion.

Instead of trajectory features, local cuboid-based features are proposed to represent the events. These features include the histograms of gradients (HOG), histograms of optical flow (HOF), and other spatio-temporal gradient features that are extracted from local 2D image patches or local 3D video cuboids. For example, based on the SIFT (Scale-Invariant Feature Transform) features, Chen et al. [28] employed MoSIFT (Motion Scale Invariant Feature Transform), which can better describe motion intensity and has stronger discriminant power. Similarly, using MoSIFT descriptors, Xu [29] extracted the low-level features of the video to detect violent events. In order to take advantage of the global spatiotemporal distribution characteristics of interest points, Bregonzio et al. [15] accumulated interest points from multiple time dimensions to form an interest point cloud as global features for behavior recognition. Using densely sampled spatio-temporal video volumes (STVs), Roshtkhari [30] create both local and global compositional graphs of volumes at each pixel to represent the event. Some examples of these features are shown in Figure 1.

However, it is quite difficult for hand-crafted-based methods to capture effective and robust behavior features due to the wide variety of monitoring scenes, complex crowd movement, and crowd density changes at any time, which can directly affect the anomaly-detection performance.

### 2.2. Deep Learning-Based Anomaly Detection Method

With the vigorous development of artificial intelligence technology, researchers began to explore detecting abnormal crowd behavior based on deep learning, which has yielded many results. Compared with the hand-crafted-based methods, the methods based on deep learning focus on extracting the high-level features of pedestrian appearance and motion in the video and can further distinguish normal behavior from abnormal behavior. These methods include the technology that is based on the Convolutional Neural Network (CNN), Auto-Encoder and Generative Adversarial Network (GAN). It can be classified into two categories: (1) using the pre-trained CNN to extract features of the video frame to represent the event and to train a detection model with a one-class classifier [13]; (2) fusing with the RNN, optical flow information or 3D-CNN [14] to learn the regularity to detect the motion and appearance anomaly. The former method [13] achieved 90.8% frame-level AUC on the UCSD ped1 dataset, while the latter achieved 85.0% frame-level AUC on the UCSD ped2 dataset. The Auto-Encoder contains an encoder and a decoder and is mainly used for data dimensionality reduction and feature extraction. Given that video clips only contain normal events, the Auto-Encoder can reconstruct the normal event with a lower error while the abnormal event is constructed with a higher reconstruction error. Furthermore, the encoder could map the normal events to latent representations, by learning a detection model such as the Gaussian Mixture Model [15,32]. This method is called the Gaussian Mixture Fully Convolutional Variational Autoencoders (GMFC-VAE) and achieves 91.2% frame-level AUC on the UCSD ped2 dataset and 83.4% frame-level AUC on the Avenue dataset. The GAN [18,33] contains a generator and a discriminator, which can capture normal data probability and can estimate the probability that a sample fits the training data distribution. They achieved 93.5% frame-level AUC on the UCSD ped2 dataset and 99% frame-level AUC on the UMN dataset. Next, the reconstruction errors of the generator or the classify result of the discriminator are used to detect anomalies. Different from the method mentioned above, the proposed method called the Convolutional Recurrent AutoEncoder (CR-AE), which is an improved form of the CNN and Auto-Encoder, can capture the irregularity of the temporal pattern and spatial irregularity, respectively. Some examples of these deep learning-based method are shown in Figure 2.

## 3. Method

Using the notation above, we formally introduce our approach in this section. We first state the anomaly detection problem formulation that we aim to deal with and then present the network architecture of the Convolutional Recurrent AutoEncoder (CR-AE).

### 3.1. Problem Formulation

The problem of the video event anomaly detection can be denoted as follows: In a video $V=\left[C_{i},i=1,\ldots ,T\right]$, where $C_{i}=\left[I_{t},I_{2},\cdots ,I_{t+k-1}\right]$ represents the video clip of the frame, $I_{t}$ and k are the length of the video frames. Here, the task is to assign each clip $C_{i}$ a binary label to indicate whether this clip contains an anomaly event ($y_{t}=1$) or not ($y_{t}=0$).

An overview of the proposed method is illustrated in Figure 3. First, the video clips that only contain the normal event are used to learn the CR-AE network as the detection model. Then, the test video clip detects the anomaly or not by the reconstruction error.

### 3.2. Learning the CR-AE Network

The CR-AE network contains a Convolutional Encoder, an attention-based ConvLSTM and a Convolutional AutoEncoder. The encoder of the AutoEncoder is composed of multiple convolutional layers. In each layer of the encoder, the model first performs a convolution operation on the original input or the output of the previous layer and outputs the result of the convolutional layer to the Covn-LSTM layer. The attention mechanism is used to obtain the current output characteristics from the hidden state of each Covn-LSTM layer. In each layer of the Convolutional Decoder, the output feature of the previous decoder layer and the output feature of the encoder are merged, and they perform the deconvolution operation. Through layer-by-layer deconvolution, the input of the original video segment is reconstructed, and the 2-norm of the input and the reconstruction result are computed as the objective function. The architecture of the proposed CR-AE network is illustrated in Figure 4.

In detail, the Convolutional Encoder encodes the input video clip. With the (l−1)-th layer feature maps $X^{t,l-1}\in {ℜ}^{n_{l-1}\times n_{l-1}\times d_{l-1}}$, the result of l-th layer can be represented as:(1)$$X^{t,l}=f\left(W^{l}*X^{t,l-1}+b^{l}\right)$$
where ∗ is the convolution operation and $f\left(\cdot \right)$ is the activation function. $W^{l}\in {ℜ}^{k_{l}\times k_{l}\times d_{l-1}\times d_{l}}$ denotes $d^{l}$ convolutional kernels of size $k_{l}\times k_{l}\times d_{l-1}$; $b^{l}\in {ℜ}^{d_{l}}$ is a bias term, and $X^{t,l}\in {ℜ}^{n_{l}\times n_{l}\times d_{l}}$ is the output feature map at l-th layer.

An attention-based ConvLSTM is adopted to capture the temporal regularity. By spanning different time steps, it can select hidden states, which are relevant to the last frames to overcome the deterioration of long-term dependencies. Furthermore, it can select relevant hidden states (feature maps) across different time steps to overcome the deterioration of the long-term dependence of the previous ConvLSTM [34]. Especially, with the l-th convolutional layer output feature $X^{t,l}\in {ℜ}^{n_{l}\times n_{l}\times d_{l}}$ of the Encoder from the previous hidden state $H^{t-1,l}\in {ℜ}^{n_{l}\times n_{l}\times d_{l}}$, the current hidden state $H^{t,l}$ is updated with $H^{t,l}=\mathrm{ConvLSTM}(X^{t,l},H^{t,l})$. Specifically, the detail of the ConvLSTM cell can be formulated as:(2)$$z^{t,l}=\sigma \left({\tilde{W}}_{XZ}^{l}*X^{t,l}+{\tilde{W}}_{HZ}^{l}*X^{t-1,l}+{\tilde{W}}_{CZ}^{k}\circ C^{t-1,l}+{\tilde{b}}_{Z}^{l}\right)$$
(3)$$r^{t,l}=\sigma \left({\tilde{W}}_{XR}^{l}*X^{t,l}+{\tilde{W}}_{HR}^{l}*H^{t-1,l}+{\tilde{W}}_{CR}^{l}\circ C^{t-1,l}+{\tilde{b}}_{R}^{l}\right)$$
(4)$$C^{t,l}=z^{t,l}\circ \tanh\left({\tilde{W}}_{XC}^{l}*X^{t,l}+{\tilde{W}}_{HC}^{l}*H^{t-1,l}+{\tilde{W}}_{CR}^{l}\circ C^{t-1,l}+{\tilde{b}}_{R}^{l}\right)+r^{t,l}\circ C^{t-1,l}$$
(5)$$o^{t,l}=\sigma \left({\tilde{W}}_{XO}^{l}*X^{t,l}+{\tilde{W}}_{HO}^{l}*H^{t-1,l}+{\tilde{W}}_{CO}^{l}\circ C^{t-1,l}+{\tilde{b}}_{O}^{l}\right)$$
(6)$$H^{t,l}=o^{t,l}\circ \tanh\left(C^{t,l}\right)$$
where ∗ and ∘ are the convolutional operator and hadamard product, respectively; σ is the activation function. ${\tilde{W}}_{XZ}^{l}$,${\tilde{W}}_{HZ}^{l}$,${\tilde{W}}_{CZ}^{k}$,${\tilde{W}}_{XR}^{l}$,${\tilde{W}}_{HR}^{l}$,${\tilde{W}}_{CR}^{l}$,${\tilde{W}}_{XC}^{l}$,${\tilde{W}}_{HC}^{l}$,${\tilde{W}}_{XO}^{l}$,${\tilde{W}}_{HO}^{l}$,${\tilde{W}}_{CR}^{l}$,${\tilde{W}}_{CO}^{l}$ are the convolution kernels and ${\tilde{b}}_{Z}^{l}$,${\tilde{b}}_{R}^{l}$,${\tilde{b}}_{R}^{l}$,${\tilde{b}}_{O}^{l}$ are the bias parameters. Different from the LSTM, all of the input, cell outputs, hidden states and gates are 3D tensors. The step length h is set as 5, and all the convolutional kernel sizes are set as the same. Next, a temporal attention mechanism is adopted to choose the relevant time steps and to obtain a refined output of feature maps $H^{t,l}$, which can be expressed by:(7)$${\overset{⌢}{H}}^{t,l}=\sum_{i\in (t-h,t)}\alpha^{i}H^{t,l}$$
(8)$$\alpha^{i}=\frac{\exp\{\frac{V(H^{t,l})V(H^{i,l})}{\lambda }\}}{\sum_{i\in (t-h,t)}\exp\{\frac{V(H^{t,l})V(H^{i,l})}{\lambda }\}}$$
where $V\left(\cdot \right)$ denotes vector and λ is a rescale factor (λ=10.0). The last hidden state $H^{t,l}$ is used as a group-level context vector, and the importance weight $\alpha^{i}$ is measured by the softmax function. In this way, the attention-based ConvLSTM can capture the irregularity of the temporal pattern and spatial irregularity.

The Convolutional Decoder is used to decode the feature map obtained in the previous step to obtain the reconstructed video clip. In detail, the Convolutional Decoder is expressed as:(9)$$X^{t,l-1}=\left\{\begin{array}{c} f\left({\tilde{W}}^{t,l},⊗,{\overset{⌢}{H}}^{t,l},+,b^{t,l}\right)\;\;,\;l=4\\ f\left({\tilde{W}}^{t,l}⊗\left({\overset{⌢}{H}}^{t,l}⊕{\overset{⌢}{X}}^{t,l}\right)+{\overset{⌢}{b}}^{t,l}\right),\;l=3,2,1 \end{array}\right.$$
where ⊗ and ⊕ are the deconvolution and concatenation operations, respectively; $f\left(\cdot \right)$ is the activation unit; ${\tilde{W}}^{t,l}$ and $b^{t,l}$ are the filter kernel and bias parameter, respectively. The reconstructed video clips from the previous layer of the decoder and the output of the previous ConvLSTM layer are combined and fed into the next deconvolution layer. The final output $X^{t,0}$ denotes the reconstructed video clip.

The detailed configurations of the proposed CR-AE model architecture are presented in Section 4.3. Finally, the objective function of the CR-AE model can be defined as the reconstruction error over the input video clips as below:(10)$$L=\sum_{k}{\left\|V_{k}-f_{W}\left(V\right)\right\|}_{2}^{2}$$
where $V_{k}$ and $f_{W}\left(V\right)$ are the video clip and reconstructed video clip.

### 3.3. Prediction

After training the model, the reconstruction error between the input frame $I_{x,y}^{i}$ and the reconstruction frame $f_{w}(I_{x,y}^{i})$ are represented as follows:(11)$$R(x,y,t)={\left\|I_{x,y}^{i}-f_{w}(I_{x,y}^{i})\right\|}_{2}$$
where $f_{w}$ is the learned CR-AE model. Then, the frame-level anomaly detection evaluation criteria can be represented by the sum of the all the pixel errors as below:(12)$$e(i)=\sum_{\left(x,y\right)}R(x,y,t)$$

Finally, the final frame-level score is
(13)$$S_{i}=\frac{e\left(i\right)-\min_{i}e\left(i\right)}{\max_{i}e\left(i\right)}$$


The scores estimated from a frame of anomalous events are expected to be higher than those for normal events, and a threshold θ is set to determine the sensitivity of the anomalous detection method.

## 4. Experiment and Results

### 4.1. Datasets

To verify the method proposed in this paper, we performed experiments on two publicly available video anomaly datasets, namely the UCSD PED2 dataset [35] and the ShanghaiTech [36] dataset. Both of the two datasets have their own challenges and unique particularity, such as abnormal events, degradation of video quality, complex background environment, etc. Therefore, the model needs to be experimented on the two datasets separately, which are briefly introduced as follows.

The UCSD dataset is a collection of footage from a stationary camera overlooking the sidewalk at 10 frames per second. In this dataset, anomaly events are caused by non-pedestrians and abnormal pedestrian movements on the sidewalk. Specifically, some abnormal examples include cyclists, skaters, cars, etc. This dataset has two different subsets, PED1 and PED2, which are divided by the working direction. This paper only adopts the second scene, UCSD PED2 for experimentation. Ped2 is parallel to the camera plane and is split into 16 training clips and 14 test clips, consisting of 4560 frames and with a resolution of 320 × 240.

The ShanghaiTech dataset is one of the largest datasets and was created to expand scene diversity. Compared to the other dataset, the ShanghaiTech dataset contains more video clips, split into 330 training and 107 test video clips, which are taken in 13 different scenes and a large number of different anomaly types. There are around 316 K video frames with a resolution of 856 × 480 in this dataset. In addition, it contains 130 abnormal events that include anomalies caused by sudden movements such as bicycles on the sidewalk, chases, and quarrels.

Figure 5 presents some examples of the two datasets.

### 4.2. Implementation Details

Before training the model, many details need to be emphasized. We first convert all frames of the video clip to a grayscale image and then resize them to 227 × 227. Five consecutive frames are used as the input of the model. In detail, the C1-C4 consists of 128 3 × 3 convolutional kernels, 64 3 × 3 convolutional kernels, 64 3 × 3 convolutional kernels, and 32 3 × 3 convolutional kernels, as well as 2 × 2, 2 × 2, 2 × 2, and 2 × 2 strides, respectively. The Decoder comprises the reverse architecture of the encoder. It contains four deconvolutional layers: DeConv1-DeConv4 with 32 3 × 3 convolutional kernels, 64 3 × 3 convolutional kernels, 64 3 × 3 convolutional kernels, and 128 3 × 3 convolutional kernels, as well as 2 × 2, 2 × 2, 2 × 2, and 2 × 2 strides, respectively. The decoder can combine different deconvolutional and ConvLSTM layers to obtain the feature maps, which effectively improve the anomaly detection performance. Detail of the CR-AE model are shown in Table 1.

The network weights are initialized by the “Xavier” method and are optimized by the Adam optimizer [37] to minimize the above loss. The Adam optimizer computes dimensional learning rates to adjust the gradient rates through all previously updated functions in each dimension. The Adam optimizer is widely used due to its strong convergence and empirically successful theory. The learning rate of the Adam optimizer is set at a learning rate of 0.0001, a weight decay of 0:9 for each 100 epochs, a hyperparameter $\beta_{1}$ of 0.9, and a hyperparameter $\beta_{2}$ of 0.999. The experiment is performed on a PC desktop with Intel Core i9-12900 CPU, NVIDIA GeForce GTX 3080 GPU and 32 GB RAM.

### 4.3. Results on the UCSD ped2 Dataset

On the UCSD ped2 dataset, we compared the results with the existing state-of-the-art methods, including the MPPCA [35], mixture dynamic texture (MDT) [35], 2D Convolutional AutoEncoder method (MT-FRCN [10], Conv2D-AE [14]), 3D Convolutional AutoEncoder method (Conv3D-AE) [14], AutoEncoder method based on Convolutional Long- and Short-term Memory Network (ConvLSTM-AE) [21], Stacked Recurrent Neural Network (StackRNN) [36], Baseline method [38], Semiparametric Scan Statistic (SSS) [39], Online GNG [40] and Unmasking [41], Appearance and Motion DeepNet (AMDN) [13]. Among these methods, the first five use handcrafted features and the last eight use deep learning techniques, the latter including common techniques such as convolutional neural networks, recurrent neural networks, autoencoders, generative adversarial networks, etc.

Frame-level evaluation criterion is adopted to evaluate the performance of the proposed method. For this criterion, the frame is determined as abnormal if at least one pixel of a frame is marked as abnormal. In order to use a frame-level criterion for evaluation, the time label is used to determine the true positive and false positive of the metric. Then, the detection rate (True Positive Rate, TPR) and false alarm rate (False Positive Rate, FPR) of the method are computed, as shown below:(14)$$TPR=\frac{TP}{TP+FN}\;$$
(15)$$FPR=\frac{FP}{FP+TN}\;$$

The receiver operating characteristics (ROC) are plotted with the true positive rate on the *y*-axis vs. the false positive rate. Then the area under the curve (AUC) is computed with different thresholds θ as the evaluation metric. A higher AUC score manifests better anomaly detection effects.

All the results of the comparison methods are taken from their respective papers. The qualitative frame-level evaluation results, in the form of ROC curves, are shown in Figure 6.

From Figure 6, it can be observed that the proposed method achieves a larger area under the curve (AUC) except for the Baseline method [19]. By visual observation, it is difficult to distinguish the size of the AUC of these two methods from the figure. The quantitative results frame-level evaluation, in the form of AUC, are presented in Table 2. It is obvious from Table 2 that the AUC of the proposed method outperforms the Baseline method, with a 0.2% frame-level AUC lead. More specifically, the performance of the deep learning methods surpasses the hand-crafted features-based method. Among the fourteen algorithms, our method obtains the best result with a 95.6% frame-level AUC.

### 4.4. Results on the ShanghaiTech Dataset

The ShanghaiTech dataset is a recently proposed dataset, has a large number of frames, and requires a relatively large calculation cost. Only a few methods have been tested on this dataset. These compared methods include the Conv2D-AE [14], StackRNN [36], Baseline [38], Asymptotic Bound [32] and MemAE [23]. The quantitative results in the form of AUC are presented in Table 3.

It is shown that the proposed CR-AE method achieved the best detection results on this dataset. Specifically, the AUC of the proposed method is 0.3% better than the Baseline [38] method. However, the method proposed obtained a 73.1% frame-level AUC on the ShanghaiTech dataset, which is much lower than the frame-level AUC obtained on the UCSD Ped2 dataset. This is mainly because the ShanghaiTech dataset is more complex and contains more challenges. It is more complex, including multiple scenes, multiple frames, and abnormal events that have not previously appeared in other datasets.

### 4.5. Visual Results

The detection results are visualized to further evaluate the performance of the proposed CR-AE model. As Figure 6 depicts, the frame-level detection results and some video screenshots of the two datasets are provided. In detail, the horizontal coordinate is the time of the video frame, the vertical coordinate abnormal score has been normalized to 1, and the red area represents the anomaly frames. It is evident that the proposed method can accurately detect video anomalies and can predict anomaly scores close to zero on normal videos, which demonstrates the effectiveness and robustness of the proposed CR-AE. Furthermore, it can be observed that the area with larger anomaly scores can correspond to the ground truth. Some abnormal events, such as bicycles and cars on the sidewalk, fights, and pushes, can basically be detected. Additionally, Figure 7 also provides some key frames of the detection results. When an abnormal event occurs suddenly, such as a car appearing on the scene in the right panel of Figure 7a, the anomaly score increases suddenly; on the contrary, if the anomalous object leaves the camera’s field of view, as shown in the left panel of Figure 7b, the anomaly score rapidly declines.

Examples of better and worse abnormality detection results are shown in Figure 8. The first row shows the examples of the better cases with a higher frame-level score and, the second row shows the examples of the worse cases with a lower frame-level score. It is obvious that the anomalies such as cars (Figure 8a–d) and bikes (Figure 8b) move on the sidewalk, and intense movements (Figure 8c) are easy to detect. However, occluded (Figure 8e–g) and poorly illuminated (Figure 8f) anomalous objects are difficult to detect. Furthermore, detecting anomalous events with little movement such as a lost package (Figure 8h) is also challenging for the proposed methods.

### 4.6. Computational Efficiency

Table 4 shows the detection speed comparison between our method and the other detection methods on the UCSD ped2 dataset, and the results of the comparison methods are from their corresponding articles. Some information, such as the computing environment and RAM, is not provided in these papers, but this does not affect the preliminary comparison of the results. The hardware environment of the whole experiment process is Intel Core i9-12900 CPU, NVIDIA GeForce GTX 3080 GPU and 32GB RAM, and the computing platform is Python 3.7 and Tensorflow 2.5. As can be seen from Table 3, the detection speed of the method proposed in this paper is 249 fps, which reaches the real-time detection speed (25 fps) and significantly exceeds the detection speed of other comparison methods.

## 5. Conclusions

In this study, we have introduced a Convolutional Recurrent AutoEncoder (CR-AE) to explicitly model the normal dynamics in video sequences for anomaly detection. The framework was able to model both spatial and temporal irregularities of the video data, which are based on the combination of an attention-based Convolutional Long–Short-Term Memory (ConvLSTM) network and the convolutional encoder of the AutoEncoder. Then, the reconstruction errors of the convolutional decoder were further employed to detect abnormal frames. Both the qualitative and quantitative results showed that the proposed method outperforms the state-of-the-art anomaly detection method on the UCSD ped2 dataset and the ShanghaiTech dataset. In the future, we will further study online and adaptive model updating to improve the performance of video anomaly detection. The limitations of the study are that our method could only provide frame-level detection results, which are unable to locate anomaly events. Another future research focus is on object-level and pixel-level anomaly detection.

## Figures and Tables

**Figure 1 sensors-22-04647-f001:**
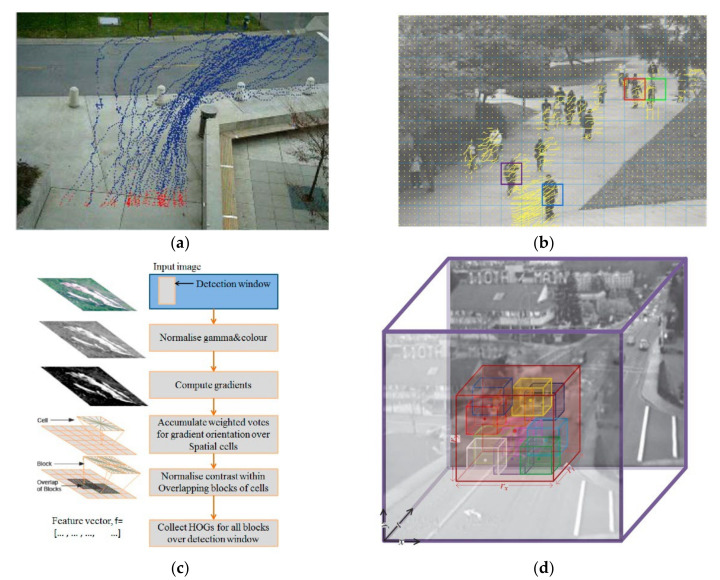
Examples of the hand-crafted feature. (**a**) Object trajectory [24]. (**b**) Dense trajectory [26]. (**c**) Histograms of gradients (HOG) [31]. (**d**) Spatio-temporal video volumes (STVs) [30].

**Figure 2 sensors-22-04647-f002:**
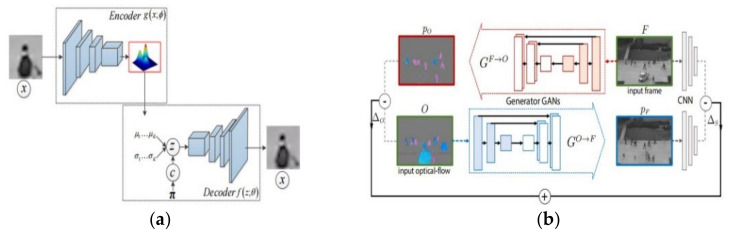
Examples of the deep learning-based method. (**a**) GMFC-VAE [32]. (**b**) GAN [18,33].

**Figure 3 sensors-22-04647-f003:**
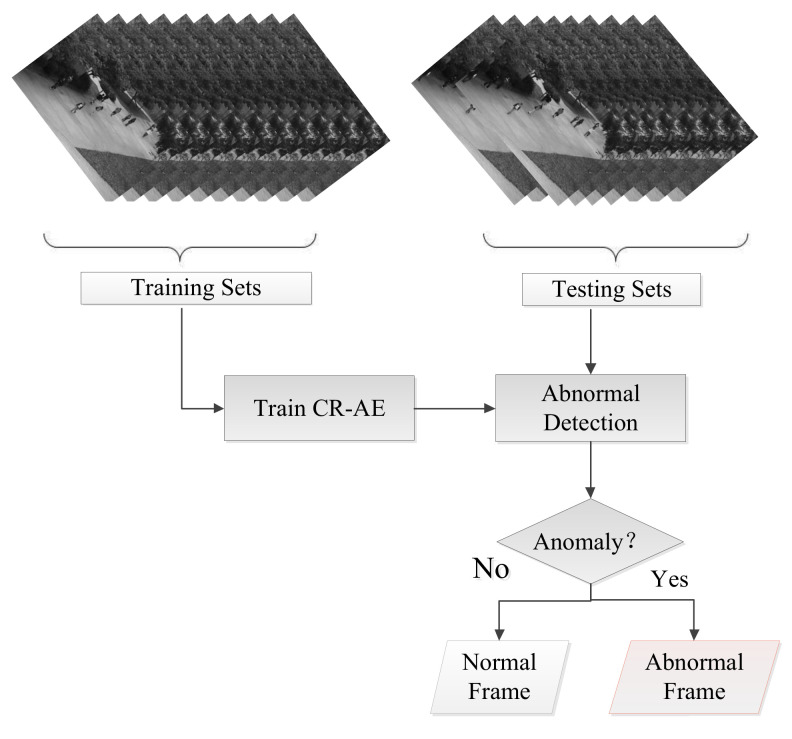
Overview of our proposed method.

**Figure 4 sensors-22-04647-f004:**
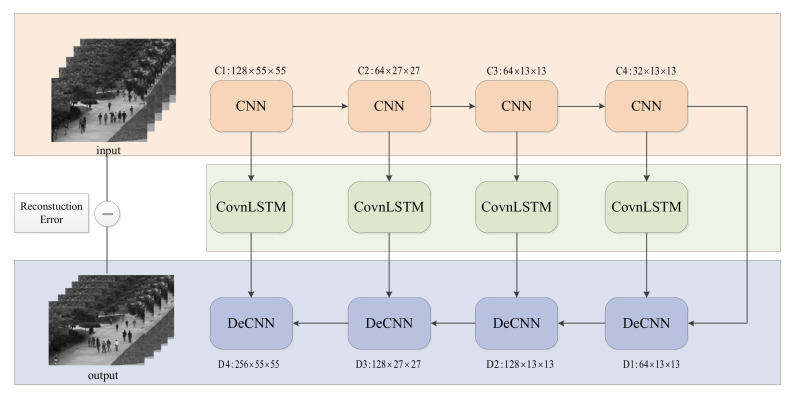
Overall architecture of the proposed CR-AE model.

**Figure 5 sensors-22-04647-f005:**
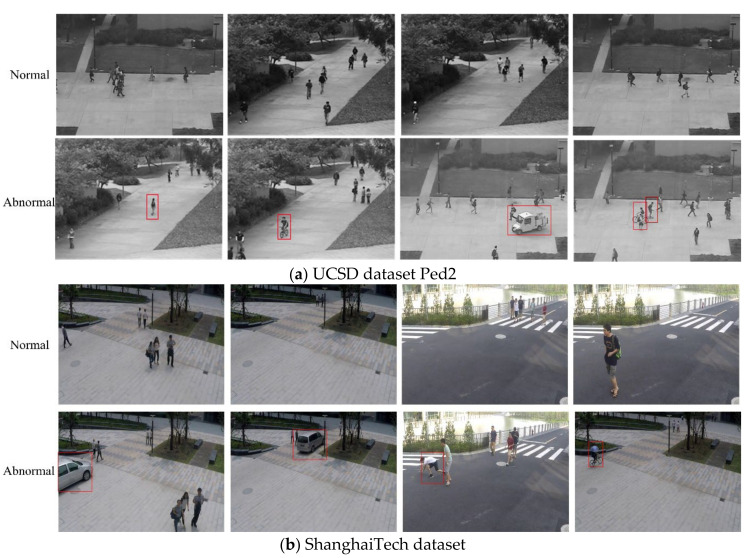
This is a figure. Schemes follow the same formatting.

**Figure 6 sensors-22-04647-f006:**
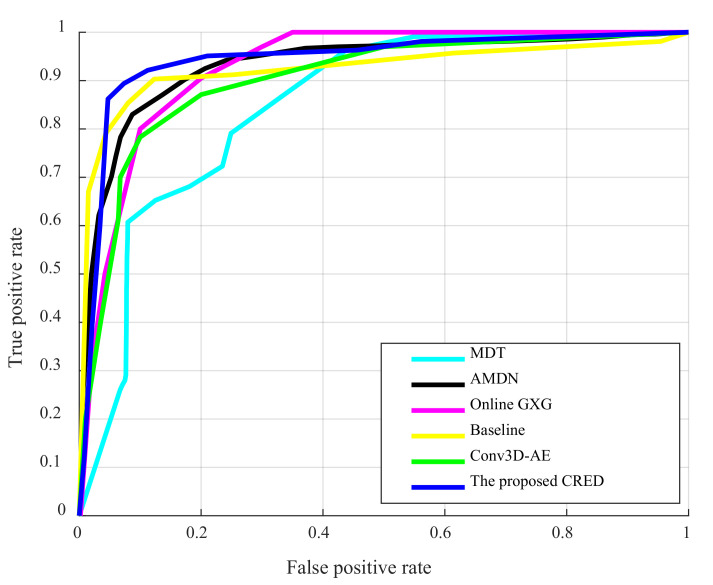
ROC curves for the UCSD Ped2 dataset.

**Figure 7 sensors-22-04647-f007:**
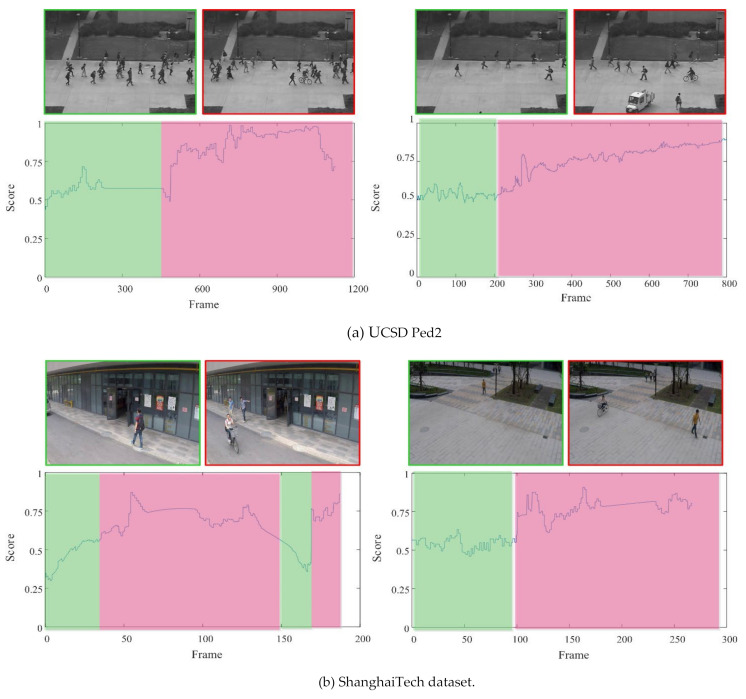
Visualization of the testing results.

**Figure 8 sensors-22-04647-f008:**
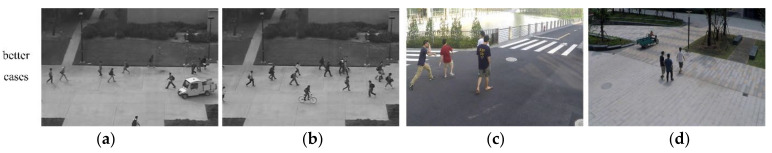
Examples of better and worse abnormality detection results. (**a**) cars on the sidewalk. (**b**) cyclists on the sidewalk. (**c**) intense movements. (**d**) cars on the sidewalk. (**e**) occluded, cyclists on the sidewalk (**f**) poorly illuminated, cyclists on the sidewalk. (**g**) occluded, scooters on the sidewalk. (**h**) lost package.

**Table 1 sensors-22-04647-t001:** Specifications of the CR-AE model.

Layer	Input	Kernel Size	Stride/ Pad	Output	Last/ Next Layer
Input	5 × 227 × 227				
Conv1	5 × 227 × 227	3 × 3	2/0	128 × 55 × 55	Input/Conv2 + Lstm1
Conv2	128 × 27 × 27	3 × 3	2/0	65 × 27 × 27	Conv 1/Conv3 + Lstm2
Conv3	64 × 27 × 27	3 × 3	2/0	64 × 13 × 13	Conv 2/Conv4 + Lstm3
Conv4	64 × 13 × 13	3 × 3	2/0	32 × 13 × 13	Conv 3/De-conv1 + Lstm4
Lstm1	128 × 55 × 55	N/A	N/A	128 × 55 × 55	Conv1/De-conv4
Lstm2	64 × 27 × 27	N/A	N/A	64 × 27 × 27	Conv2/De-conv3
Lstm3	64 × 13 × 13	N/A	N/A	64 × 13 × 13	Conv3/De-conv2
Lstm4	32 × 13 × 13	N/A	N/A	32 × 13 × 13	Conv4/De-conv1
De-conv1	32 × 13 × 13	3 × 3	2/0	64 × 13 × 13	Lstm4 + Conv4/De-conv2
De-conv2	64 × 13 × 13	3 × 3	2/0	128 × 27 × 27	Lstm3 + Conv1/De-conv3
De-conv3	128 × 27 × 27	3 × 3	2/0	256 × 55 × 55	Lstm2 + Conv2/De-conv4
De-conv4	128 × 55 × 55	3 × 3	2/0	5 × 277 × 277	Lstm3 + De-conv3/Output
Output	5 × 277 × 277				

Input, input layer; Conv, convolutional layer; Lstm, ConvLSTM layer; De-conv, deconvolutional layer; Output, output layer. The Encoder and Decoder consist of Conv1, Conv2, Conv3, Conv4 and De-conv1, De-conv2, De-conv3, De-conv4, respectively.

**Table 2 sensors-22-04647-t002:** Comparison with the state-of-the-art methods in terms of AUC% on the USCD Ped2 Dataset.

Method	AUC
MPPCA [35]	69.3%
MDT [35]	82.9%
SSS [39]	94.0%
Online GNG [40]	94.0%
Unmasking [41]	82.2%
ADMN [13]	90.8%
MT-FRCN [10]	92.2%
Conv2D-AE [14]	85.0%
Conv3D-AE [14]	91.2%
ConvLSTM-AE [21]	88.1%
StackRNN [36]	92.2%
Baseline [38]	95.4%
The proposed CR-AE	95.6%

**Table 3 sensors-22-04647-t003:** Comparison with the state-of-the-art methods in terms of AUC% on the ShanghaiTech dataset.

Method	AUC
Conv2D-AE [14]	60.9%
StackRNN [36]	68.0%
Baseline [38]	72.8%
Asymptotic Bound [32]	70.9%
MemAE [23]	71.2%
The proposed CR-AE	73.1%

**Table 4 sensors-22-04647-t004:** Running time comparison of the UCSD Ped2 dataset.

Method	Computing Environment	CPU	GPU	RAM	Detection Speed (fps)
MDT [35]	-	3.0 GHz	-	2.0 GB	0.04
StackRNN [36]	Python + Tensorflow	3.5 GHz	-	16 GB	120
AMDN [13]	MATLAB 2015	2.1 GHz	Nvidia Quadro K4000	32 GB	0.11
Unmasking [41]	Python + Tensorflow	-	GTX TITAN Xp	-	20
Proposed CR-AE	Python 3.7 + Tensorflow2.5	5.1 GHz	NVIDIA GTX 3080	32 GB	249

## Data Availability

Not applicable.

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
