# Peer review of "Video Anomaly Detection Based on Convolutional Recurrent AutoEncoder"

_sensors, 2022, doi:10.3390/s22124647_

Round 1

Reviewer 1 Report

The paper is well structured, and it proposes a new deep learning-based method called the Convolutional Recurrent AutoEncoder (CR- AE) for video anomaly detection. Different from other works, the proposed method could simultaneously detect the spatial and temporal anomaly.

  1. In the Introduction you mentioned “That is due to the reason that if the CR-AE has never observed a similar pattern before, it may not be able to reconstruct the input video clip well.” Is it positive or negative? I don’t understand the context and the effects of this phrase.
  2. Despite the contributions are explicitly numerated, it doesn’t indicate the impact of the study:

-the first one mentions the methodology, but how is it innovative?

-the second contribution should include the advantages of using only one net vs. the conventional two-stream network.

  1. The values in Table 2 and Table 3 don’t include the confidence interval or information about deviation, then the difference could be not significant. It is not possible to state that your result is better than the other proposal (really 95.6% is significantly better than 95.45?)
  2. The results should be compared with recent works and publications in high impact journals, like:

- X. Zhang, S. Yang, J. Zhang, W. Zhang, Video anomaly detection and localization using motion-field shape description and homogeneity testing, Pattern Recogn. 105 (2020) 107394.

-Romany F. Mansour, José Escorcia-Gutierrez, Margarita Gamarra, Jair A. Villanueva, Nallig Leal, Intelligent video anomaly detection and classification using faster RCNN with deep reinforcement learning model, Image and Vision Computing, Volume 112, 2021, 104229, https://doi.org/10.1016/j.imavis.2021.104229.

-B.S.Murugan,M. Elhoseny, K. Shankar, J. Uthayakumar, Region-based scalable smart system for anomaly detection in pedestrian walkways, Computers & Electrical Engineering, 75 (2019) 146–160.

  1. English must be improved.

Reviewer 2 Report

The paper presents an interesting subject. It is well written. The following aspects must be presented more clearly:

-section Related work must contain also obtained results

-the novelty of the proposed method must be presented in correlation with other existing ones

-results must be described in more details: what are the best cases (what are the bad ones), did illumination conditions influence the obtained results?

-how it is made the difference between anomaly and not-anomaly in case that many cars appear in the image (eg. for figure 5a)

-uniformise the usage of Fig. (eg. fig. 4 - line 334)  and figure 

Reviewer 3 Report

Dear Authors,

   Thanks so much for your manuscript submission to special issue on MDPI Journal of Sensors. This short research article is comprehensively good, while the organization of this paper still needs some adjustment, the use of English on this research article is acceptable, in spite that some subsections need improvement. Therefore, I think that some edits to improve the major and minor problematic issues are necessary, after addressing all reviewers' comments as suggested, it may enter the double-decision process.

   Major problematic issues need to be addressed in your revision:

   a) Abstract: The length of current version is acceptable, however, a few aspects need improvements. The first sentence contains grammatical error, which is also redundant. The following several sentences can be expanded with technical details. The presence of your technical approach need to be a bit more specific, and the concluding remarks should include the keynote quantitative results related to your experimental study. The overal length after updating are suggested to be within 180~200 words. Thanks a lot! 

   b) Introduction: The structure of current session generally looks fine, while the use of English should be improved. For instance, I believe the first passage need a major rewrite, due to no citations and lacking in specificity. i.e., why and how did video anomaly detection (VAD) become a major challenge in the computer vision field? Why conducting research? What are the possible applications of involved research study? Which are the classical representative works in this area? Also, I think the authors should learn to apply connection sentences and help cohesively distributing the required information. In Lines 32-58, the presence of feature extraction and model learning should also be extended with crucial details of typical approaches.
   Besides, in Lines 89-98, the main summary on contributions of this paper, need to supplement with more specific details (the current version is quite generic) in each manifold. Also, I think the last paragraph is good enough to summarize the organization on the rest sections of this paper.

   c) Section 2 (Related Work): This section looks just fine but contains too much simple narrations. I have some advice for the co-authors that you may select most representative architecture of the related technical approaches, which might help clarify the visual understandings on each topic. Besides, a table for presenting the tabulated previous studies, can be an alternative plan to make summary on the related work. Thanks a lot!

   d) Method (Section 3): In general, it looks fine for the overview and overall architectures. There are still room for further updates on the narrations. Take the Line 203 and 223 as an example: what did the term "Then" make sense? Similar issues such as "That is" at Line 219, should be fixed. Meanwhile, the blanking issue at Line 209, the formatting issues such as the capital letter "Where" at Line 219, the zero character indencing of "where" at Lines 200, 212, 219, 226, 234 and 239, must be calibrated. In addition, I suggest the authors avoid using too many hard conjuctions such as "thus", "then", "is proposed", etc. Do your best to improve the comprehensive quality of literal statements in a few subsections, at least just making them more readable.

   e) Figures and Tables: Most of the tables are fine, while the title of Table 4 at Line 391 should be shifted to the next page (don't let one table or figure crossing over two pages). Besides, the interval before and after each figure and table should be placed with 12 pts (similar as the MDPI template did).  The visual quality of each figure (size and image resolution) are acceptable. The font type of each figure (as well as the legends) should be uniform, either using Times New Roman or Palatino Linotype. Regarding formats, please apply middle-alignment for each of the headings. Thanks a lot!

   f) Section 4: I think this section can be named as "Experiments and Results" while the "Discussions" could be possibly shifted into the discussion section, if applicable. I suggest the authors using the evaluation metrics directly applied on your study.  PS: Regarding the test results, can you explain why the improvement on AUC score (%) of your proposed method is close to <5% improvements to StackRNN [21], Baseline [18], Asymptotic Bound [27] and MemAE [28] for ShanghaiTech dataset?  Why the Ablation study or sensitivity analysis is missing in your quantitative analysis? Do you have further evidence to demonstrate the applicability of your proposed framework when crossing over various datasets? Please briefly explain.

   g) Discussions: I may suggest the authors adding a discussion section for presenting some further qualitative analysis, i.e., the limitations of study, some parallel comparison of your proposed work towards the mainstream of latest research topics, and further comments (if any) on the quantitative scores related to performance evaluation. Please apply the required edits.

   h) Section 5: This section had better be entitled with "Conclusions and Future Work". The first paragraph should be filled with some keynote quantitative scores on concluding remarks, and be a bit more specific on what degree of improvement, what are the most representative advantages of shortcomings on your proposed framework? While this paragraph shows a single sentence claiming future work, it is supposed to apply the specified summary of research challenges, and the orientations of prospective study should be a little bit more specific. Please re-arrange this part. Thanks a lot!

   i) References: A few problematic aspects should be addressed. (i) Apply the required abbreviated, italic formats on the title of journals when citing, i.e., "transactions" --> "Trans.". (ii) Fill in the missed information (volume and page numbers) for each conference proceedings and calibrate the citation style. iii) I suggest the authors proceed to review and cite both conventional, newer and latest approaches, in one decade range, especially these paper published in latest three years 2019-2022 which are similar / parallel to your study, can be further enhanced in your upgraded version. These updates will make your citations look even more stronger. (iii) Comply with current MDPI template for other tutorial formats in a list of References at various sources.

   Other minor issues suggested for your edits are listed as below:

   a) To be frank, the literal quality of this research article can be further improved. Quite a few subsections such as introduction, related work and conclusion contain Chinglish and hard conjuctions. I suggest the authors on inviting a native English speaker to carefully polish the writing aspects of this manuscript, which should include grammatical checks and proofreading in your edits before uploading the second version. Thanks very much!

   b) Please eliminate a couple of minor hyphenating in the context. When you are using MS word or Latex, please avoid hyphenating a word (which currently appears multiple time at the end of some lines to cross-over two adjacent lines). The MDPI online template has the options to adjust that.

   c) In some subsections, quite a few "half-spacing" intervals between an equation "=" or two adjacent sentences (as well as a bracket followed by the start of a sentence) are missing. Fix each occurrence when proofreading.

   d) Align the location of each figure and the statements below. Be sure that the size and position of figures and tables comply with the MDPI template.

   Once again, many thanks and we look foward to reviewing your upgraded version coming into double-decision process, and wish you the best of luck for paper acceptance at MDPI Journal of Sensors. Stay well and take care!

Best wishes,

Yours faithfully,

Round 2

Reviewer 2 Report

Since all my comments were addressed, I recommend to publish the paper.